# The Multivalent Polyampholyte Domain of Nst1, a P-Body-Associated *Saccharomyces cerevisiae* Protein, Provides a Platform for Interacting with P-Body Components

**DOI:** 10.3390/ijms23137380

**Published:** 2022-07-02

**Authors:** Yoon-Jeong Choi, Yujin Lee, Yuxi Lin, Yunseok Heo, Young-Ho Lee, Kiwon Song

**Affiliations:** 1Department of Biochemistry, College of Life Science and Biotechnology, Yonsei University, Seoul 03722, Korea; yoon3004@yonsei.ac.kr (Y.-J.C.); 2019162016@yonsei.ac.kr (Y.L.); 2Research Center for Bioconvergence Analysis, Korea Basic Science Institute (KBSI), Chungbuk 28119, Korea; linyuxi@kbsi.re.kr (Y.L.); uoonsek1@kbsi.re.kr (Y.H.); mr0505@kbsi.re.kr (Y.-H.L.); 3Department of Bio-Analytical Science, University of Science and Technology (UST), Daejeon 34113, Korea; 4Graduate School of Analytical Science and Technology (GRAST), Chungnam National University (CNU), Daejeon 34134, Korea

**Keywords:** P-body, liquid–liquid phase separation, Nst1, polyampholyte domain, aggregation-prone domain, *Saccharomyces cerevisiae*

## Abstract

The condensation of nuclear promyelocytic leukemia bodies, cytoplasmic P-granules, P-bodies (PBs), and stress granules is reversible and dynamic via liquid–liquid phase separation. Although each condensate comprises hundreds of proteins with promiscuous interactions, a few key scaffold proteins are required. Essential scaffold domain sequence elements, such as poly-Q, low-complexity regions, oligomerizing domains, and RNA-binding domains, have been evaluated to understand their roles in biomolecular condensation processes. However, the underlying mechanisms remain unclear. We analyzed Nst1, a PB-associated protein that can intrinsically induce PB component condensations when overexpressed. Various Nst1 domain deletion mutants with unique sequence distributions, including intrinsically disordered regions (IDRs) and aggregation-prone regions, were constructed based on structural predictions. The overexpression of Nst1 deletion mutants lacking the aggregation-prone domain (APD) significantly inhibited self-condensation, implicating APD as an oligomerizing domain promoting self-condensation. Remarkably, cells overexpressing the Nst1 deletion mutant of the polyampholyte domain (PD) in the IDR region (Nst1_∆PD_) rarely accumulate endogenous enhanced green fluorescent protein (EGFP)-tagged Dcp2. However, Nst1_∆PD_ formed self-condensates, suggesting that Nst1 requires PD to interact with Dcp2, regardless of its self-condensation. In Nst1_∆PD_-overexpressing cells treated with cycloheximide (CHX), Dcp2, Xrn1, Dhh1, and Edc3 had significantly diminished condensation compared to those in CHX-treated Nst1-overexpressing cells. These observations suggest that the PD of the IDR in Nst1 functions as a hub domain interacting with other PB components.

## 1. Introduction

The phenomenon of biomolecular phase separation has expanded our understanding of biomolecular condensation in cells [1]. Biomolecular condensates include many non-membranous cellular structures, such as Cajal bodies, nuclear speckles, histone-locus bodies, promyelocytic leukemia (PML) nuclear bodies (NB) in the nucleus [2,3,4,5], P-bodies (PBs), stress granules (SGs), and germ granules in the cytoplasm [1,2,4,6,7,8,9]. These membrane-less cellular structures are not random biomolecule mixtures. Some components are shared in different condensates, but each membraneless organelle contains a specific group of proteins and RNA/DNA, differentiating it from others. Not all of the condensate components are critical for inducing condensation, but a few components, so-called scaffolds, play crucial roles [10].

Ribonucleoprotein (RNP) granules are among the most representative biomolecular condensates and are an efficient model for studying biomolecular condensation in cells. RNA, the general component of these condensates, is essential for these condensation processes [11]. RNase treatment disperses isolated messenger RNPs (mRNPs) in vitro [11]. Additionally, decreased free ribosomal mRNA influx alleviates mRNP granule condensation in cells [12]. These phenomena strongly support the notion that RNA is a critical factor for molecular condensation.

mRNP granules also contain hundreds of proteins [10,13,14,15,16,17]. Scaffold proteins that function as nodes for protein–protein interaction networks are typically sufficient to form condensates [10]. Scaffold proteins have the intrinsic potential to induce condensation, while client elements are concentrated within the structure often by direct interactions with scaffolds but are not required for condensate formation [18]. The scaffold proteins show a few distinctive characteristics that distinguish them from client proteins. First, numerous scaffold proteins exhibit self-oligomerizing properties. The RING finger-B box-coiled coil (RBCC) motif [19,20,21] contains an N-terminal RING, B1-box, B2-box, and a C-terminal coiled-coil (CC) domain and is considered essential for PML oligomerization to form PML NBs [22,23,24]. In the case of Ras-GTPase-activating protein (SH3 domain)-binding protein (G3BP), the dimerization domain nuclear transport factor 2 (NTF2) is insufficient but necessary for SG formation [25].

Scaffold proteins also have multivalent regions that participate in weak interactions with numerous binding partners [26,27]. Intrinsically disordered regions (IDRs) with few three-dimensional (3D) structures and little specificity [28,29,30,31,32,33] are reportedly necessary for liquid–liquid phase separation (LLPS) dynamics and multivalency. IDRs of the heterogeneous RNP family, including hbRNPA1 in SGs [31], hbRNPA2B1 [33], RNA helicase Ddx4 in nuage [28], and Laf-1 in P-granules [34,35], are sufficient to mediate phase separation. Low-complexity domains (LCDs) [33,36] such as the poly-Q/N prion-like domain (PrD) and the arginine-glycine-rich (RGG) motif [4,34,37] are also known as critical modifiers for generating LLPS. The polyampholyte or polyelectrolyte region of the IDR may function as a sticker to promote LLPS [34]. Although previous research has established the link between IDR and multivalency, it has not elucidated the syntax of molecular condensation.

PBs of the budding yeast *Saccharomyces cerevisiae* provide an excellent system for studying the elements and mechanisms to form cellular condensates. The predominant components of yeast PBs are mRNA decapping protein Dcp1 and Dcp2, which constitute the decapping enzyme, enhancer of mRNA decapping protein 3 (Edc3), Pat1, Dhh1, and the Lsm1-7 complex, all of which are mRNA-binding proteins that stimulate mRNA decapping [37,38,39]. Predominantly, multivalent Edc3 interactions appear to drive PB formation. Edc3 serves as a scaffold for PB assembly, primarily under glucose deprivation when PB formation is robust. In a previous study, we identified Nst1 as a novel PB component. Nst1 accumulates in PBs more densely in stationary phase cells and under glucose deprivation. Ectopically overexpressed Nst1 is self-condensed and induces the condensation of other PB components, such as Dcp2, indicating that Nst1 has the intrinsic potential to self-condensate and accumulate other PB components [40].

Here, we dissected Nst1 by overexpressing various Nst1 domain deletion mutants to understand the functions of distinctive Nst1 sequence elements in its self-condensation and recruitment of other PB components and improve our knowledge of molecular condensation in cells.

## 2. Results

### 2.1. The Nst1 C-Terminal Domain (CTD) Contains Polyampholyte and Aggregation-Prone Regions

We previously reported that Nst1, similarly to Edc3, induced Dcp2 accumulation via self-condensation and physical interactions with other PB components [40]. These observations strongly suggest that Nst1 contains an oligomerizing domain similar to Edc3, with the intrinsic potential to drive self-condensation. Nst1 is a 141 kDa protein consisting of 1240 amino acids with a unique sequence distribution (Appendix A). To determine the properties of Nst1 in self-generating condensates and the induced condensation of other PB components, we analyzed the Nst1 sequence using multifaceted sequence prediction tools: protein structure prediction using GalaxyWEB (http://galaxy.seoklab.org/, accessed on 12 October 2018) (Appendix A), IDR prediction with IUPRED2A, PONDR, and DISOPRED3, and aggregation-prone region prediction with AGGRESCAN, Tango, and PASTA 2.0 (Figure 1A,B).

The 980-amino acid sequence (from amino acid 131 to 1110), excluding 130 amino acids of each Nst1 N- and C-terminus, was analyzed because of the 1000 amino acid limit of GalaxyWEB (Appendix A). N-terminus (residues 1–429) (data not shown) and C-terminus (residues 430–1240) (Appendix AC) structures were also predicted independently. Nst1 was expected to be low-ordered and to not form a globular 3D structure (Appendix A). Based on the prediction of the secondary structure by GalaxyWEB, Nst1 could be divided mainly into two domains: the N-terminal domain (NTD) (1–406) and CTD (430–1240), with a short 23-amino acid unstructured region (UR) between them (Figure 1C).

The 225 amino acids (residues 1016–1240) in the Nst1 C-terminus contain particularly high scores in aggregation propensity prediction (Figure 1B). Considering the aggregation propensity and protein secondary structure predictions, we designated this region as an aggregation-prone domain (APD) (Figure 1C and Appendix A).

We found that amino acids 1–32 in the NTD and 491–980 in the CTD scored highly in all three IDR predictions (Figure 1A). The polyampholyte sequence, including charged amino acid clusters D, E, R, and K with sparse hydrophobic amino acid L, was embedded in the predicted IDR (Figure 1A and Appendix A). Considering that the polyampholytic sequence was predicted as coiled-coil (CC) helices in the secondary structure prediction by GalaxyWEB, we designated this predicted region as the polyampholyte domain (PD) (Appendix A).

Polyampholyte sequences are commonly present in many IDRs [48]. Charged amino acids, such as D, E, K, and R, function as inter- and intra-molecular stickers to generate LLPS [34,49]. The molecular conformation of proteins can be deduced based on the fraction of the charged amino acids in the Das–Pappu diagram [50]. Fundamentally, the fraction of charged residues (FCR) and the net charge per residue (NCPR) determine the four regions, R1, R2, R3, and R4, in the Das–Pappu diagram. The proteins showing an FCR value smaller than 0.35 were assigned to R1 or R2. The protein sequences presented in R1 and R2 were expected to have a globular conformation of weak polyampholytes and an alternative globular conformation of a context-dependent polyampholyte, respectively. Proteins with an FCR value greater than 0.35 were classified as R3 or R4. These protein sequences were strong polyampholytes or polyelectrolytes that were expected to be coiled. We projected the sequence of Nst1 and each Nst1 deletion mutant of the predicted domain onto the Das–Pappu diagram (Figure 1D). Full-length Nst1 (1) was projected in R2, where the fraction of negatively or positively charged residues was between 0.25 and 0.35. The zone of the context-dependent polyampholyte indicates that Nst1 may not have a stable globular protein structure, and its composition may be altered in a context-dependent manner. Additionally, the Nst1 N-terminal (Nst1_NTD_) (2) and Nst1 C-terminal (Nst1_CTD_) (3) projections were close to the full-length Nst1 in the same R2 region of the diagram. This indicated that the ratio of the charged Nst1_NTD_ and Nst1_CTD_ residues was similar to that of full-length Nst1. However, as expected due to its IDR predictions, Nst1_ΔPD_ (4), the mutant lacking the polyampholyte region, was projected on the border of R2 and R1, in which the fraction of negatively or positively charged residues was below 0.25. This Nst1_ΔPD_ prediction indicated that deleting the Nst1 polyampholyte region could severely alter the full-length Nst1 FCR. In contrast, Nst1_ΔAPD_ (5) was projected onto the R3 zone, demonstrating that deleting the aggregation-prone domain (APD) increased the Nst1 FCR. The Nst1_ΔPDΔAPD_ projection (6) showed the offset effect of PD and APD deletions. Collectively, these predictions suggest that the unique sequence distribution of Nst1, especially the PD and APD, may enhance Nst1 self-condensation and the condensation of other PB components.

### 2.2. The Nst1 CTD Is Sufficient for Nst1 Self-Condensation

To identify the specific regions of Nst1 responsible for the self-condensation and condensation of other PB components, we designed various Nst1 domain deletion mutants with different domain combinations based on the predictions (Figure 1). Each green fluorescent protein (GFP)-tagged Nst1 domain deletion mutant was overexpressed under the galactose inducible (*GAL*) promoter of pMW20 in the wild-type cells, and its expression was confirmed by Western blot analysis (Appendix A). As reported in a previous study [40], overexpressed GFP-tagged Nst1 formed bright puncta (Figure 2A). The Nst1 mutant, Nst1_NTD_, was completely dispersed throughout the cytoplasm when overexpressed (Figure 2A). In contrast, Nst1_CTD_ formed clear puncta upon overexpression (Figure 2A). These observations demonstrate that the Nst1 CTD was sufficient to form self-condensates upon overexpression (Figure 2A).

The puncta formed by Nst1 overexpression were closely correlated with the physical LLPS properties obtained via 1,6-hexanediol treatment [27]. 1,6-hexanediol is reported to eradicate the nuclear pore permeability barrier by interfering with hydrophobic interactions in the pores and is generally used to interfere with the integrity of reversible condensates with liquid-like properties [51,52]. In budding yeast cells, treatment with 5–10% 1,6-hexanediol for 30 min can impede PB integrity but cannot disperse irreversible amyloids [27]. When cells were treated with 1,6-hexanediol, we observed that condensates of overexpressed GFP-tagged Nst1_CTD_ dispersed as those of full-length Nst1, exhibiting the liquid-like property of both condensates (Figure 2A).

Previously, we demonstrated that the accumulated GFP-tagged Nst1 condensates co-localized with endogenous Dcp2-mKate2 [40]. To quantitatively investigate the correlation between Nst1 self-condensation and its association with PBs, we analyzed the co-localization of each overexpressed Nst1 domain deletion mutant and endogenous Dcp2 (a PB marker) using van Steensel’s cross-correlation function (CCF) (Figure 2B). Here, van Steensel’s CCF determines the degree of co-localization between two different signals (red and green) by crossing the Pearson coefficients of each image signal [53]. Endogenous Dcp2-mKate2 was captured for analysis in wild-type cells whose chromosomal *DCP2* was tagged with mKate2 after each GFP-tagged Nst1 mutant was overexpressed. Van Steensel’s CCF of overexpressed GFP-tagged Nst1_NTD_ did not show a bell-shaped curve with Dcp2-mKate2, indicating that the red and green signals did not overlap (Figure 2B). However, van Steensel’s CCF of overexpressed GFP-tagged Nst1 and Nst1_CTD_ for Dcp2-mKate2 showed a bell-shaped curve, although GFP-tagged Nst1_CTD_ and Dcp2-mKate2 showed a weaker correlation than the wild-type Nst1 for Dcp2-mKate2 (Figure 2B). GFP-tagged Nst1_CTD_ and Dcp2-mKate2 correlated more closely than GFP-tagged Nst1_NTD_ and Dcp2-mKate2 (Figure 2B). These data suggest that Nst1 self-condensation is correlated with the accumulation of the Dcp2 PB marker.

To demonstrate the Nst1 domain responsible for EGFP-Dcp2 condensation, we monitored endogenous EGFP-Dcp2 in cells overexpressing Nst1, Nst1_NTD_, and Nst1_CTD_. As expected, Nst1_NTD_ overexpression did not increase EGFP-Dcp2 condensation, whereas overexpression of full-length Nst1 induced EGFP-Dcp2 condensation (Figure 2C). EGFP-Dcp2 condensation induced by Nst1_CTD_ overexpression was enhanced compared to the vector control and overexpressed Nst1_NTD_ (Figure 2C). However, EGFP-Dcp2 condensation in cells overexpressing Nst1_CTD_ was reduced compared with that in cells overexpressing full-length Nst1 (Figure 2C). To quantify the degree of EGFP-Dcp2 puncta generated, we segmented pixels of the top 0.05% intensity for puncta analysis, and the maximum intensities of the segmented puncta scaled from 0–255 were analyzed using a boxplot. The measuring method is described in detail in the Materials and Methods section and our previous study [40]. Consistent with Figure 2C, full-length Nst1 and Nst1_CTD_ overexpression increased the maximum intensities of EGFP-Dcp2 condensates, while Nst1_NTD_ overexpression did not (Figure 2D). Instead, the EGFP-Dcp2 condensates were decreased in Nst1_NTD_-overexpressing cells compared to the vector control cells (Figure 2C,D). The endogenous EGFP-Dcp2 expression level in each mutant overexpressing cell was monitored by Western blotting to confirm that the overexpression of each mutant did not affect EGFP-Dcp2 expression levels (Appendix A).

These data suggest that an intrinsic sequence factor responsible for Nst1 self-condensation is present in the Nst1 CTD. In addition, the condensation of PB components in the Nst1-overexpressed cells was produced based on Nst1 self-condensation through LLPS.

### 2.3. The APD in the Nst1 CTD Is Insufficient but Crucial for Inducing Nst1 Self-Condensation

Next, we investigated whether the predicted APD in the C-terminus induced Nst1 condensation. We constructed GFP-tagged Nst1_ΔAPD_ and Nst1_CTDΔAPD_ mutants and compared their overexpression phenotypes with those of GFP-tagged wild-type Nst1 and Nst1_CTD_ in BY4741 wild-type cells. Overexpression of GFP-tagged Nst1 and Nst1_CTD_ generated self-condensation (Figure 3A). However, when Nst1_∆APD_ was overexpressed, major GFP signals were dispersed throughout the cytoplasm. We observed the same dispersed phenotype in cells overexpressing GFP-tagged Nst1_CTD__∆APD_, where the APD was deleted in Nst1_CTD_, although Nst1_CTD_ displayed discrete puncta (Figure 3A). These observations suggest that the APD plays a critical role in Nst1 condensation. To further test the sufficiency of the APD inducing condensation, only the APD was overexpressed. This did not result in the assembly of any condensates (Figure 3A). These observations demonstrate that the Nst1 APD is crucial but insufficient for Nst1 condensation to form condensates. Consistently, the CCF of overexpressed GFP-tagged Nst1_ΔAPD_ and Nst1_CTDΔAPD_ versus endogenous Dcp2-mKate2 did not show a bell-shaped curve (Figure 3B).

In monitoring EGFP-Dcp2 in cells overexpressing these domain deletion mutants, Nst1_∆APD_ overexpression did not show EGFP-Dcp2 condensate accumulation, while overexpression of the full-length Nst1 induced EGFP-Dcp2 condensation (Figure 3C), as expected from van Steensel’s CCF. In the quantitative analysis of the EGFP-Dcp2 condensates, the overexpression of full-length Nst1 and Nst1_CTD_ increased the maximal intensities of EGFP-Dcp2 condensates compared to the vector control, while the overexpression of Nst1_∆__APD_ and Nst1_CTD__∆APD_ canceled out the effect (Figure 3D). The maximal intensities of the EGFP-Dcp2 condensates were reduced in both Nst1_ΔAPD_- and Nst1_CTDΔAPD_-overexpressing cells compared to those of the vector control cells (Figure 3D). Endogenous EGFP-Dcp2 did not appear as puncta in cells overexpressing Nst1_ΔAPD_, suggesting that the APD of Nst1 alone was unable to induce self-aggregation. Endogenous EGFP-Dcp2 expression levels in cells overexpressing each mutant were monitored by Western blotting to confirm that the overexpression of each mutant did not affect EGFP-Dcp2 expression levels (Appendix A). These data demonstrate that the Nst1 APD is the critical region for inducing Nst1 self-condensation but functions in a context-dependent manner.

### 2.4. The Nst1 PD Is Not a Critical Component in Self-Condensation but Is Responsible for Inducing Dcp2 Condensation

The polyampholyte region of proteins is a representative IDR and is anticipated to be closely related to biomolecular condensation [48,49]. We attempted to demonstrate the function of the PD in Nst1 self-condensation because an obvious polyampholyte region is present in the Nst1 CTD. Considering previous reports on the function of the polyampholyte region in LLPS [48,49], we expected that PD deletion in various Nst1 domain mutants would negatively affect self-condensate generation. We compared the punctum formation of the GFP-tagged PD deletion mutants with that of the GFP-tagged full-length Nst1 and Nst1_CTD_ upon overexpression. Unexpectedly, GFP-tagged Nst1_∆PD_ generated condensates similar to wild-type Nst1 when overexpressed, demonstrating that PD does not control Nst1 self-condensation (Figure 4A). Nst1_CTD∆PD_ overexpression also formed puncta (Figure 4A). However, the size and intensity of puncta in cells overexpressing GFP-tagged Nst1_CTD∆PD_ were reduced, compared with those in cells overexpressing GFP-tagged Nst1_CTD_ (Figure 4A), suggesting that the PD in Nst1 may only partially contribute to Nst1 self-condensation. Both condensates induced by GFP-tagged Nst1_∆PD_ and Nst1_CTD∆PD_ overexpression were dispersed in the cytoplasm in 1,6-hexanediol-treated cells, indicating their liquid-like properties (Figure 4A).

We anticipated that overexpressed Nst1_∆PD_ would induce EGFP-Dcp2 condensates and colocalize with the Nst1 overexpression because Nst1 PD deletion did not interrupt Nst1 self-condensation upon overexpression. However, in van Steensel’s CCF diagram, the localization of endogenous Dcp2-mKate2 tended to be less correlated with overexpressed GFP-tagged Nst1_∆PD_ localization than with overexpressed GFP-tagged full-length Nst1 and Nst1_CTD_ (Figure 4B).

To examine the functional potential of PD deletion in EGFP-Dcp2 condensation, we investigated EGFP-Dcp2 condensation in cells overexpressing Nst1_∆PD_. Endogenous EGFP-Dcp2 was monitored in cells overexpressing each Nst1 domain deletion mutant (Figure 4A). Unexpectedly, the intensity of EGFP-Dcp2 puncta hardly increased in cells overexpressing Nst1_∆PD_ (Figure 4C,D), although we observed that GFP-tagged Nst1_∆PD_ overexpression generated bright puncta via self-condensation (Figure 4A). Endogenous EGFP-Dcp2 expression levels in Nst1 mutant-overexpressing cells monitored by Western blotting indicated that Nst1_∆PD_ mutant overexpression did not affect EGFP-Dcp2 expression levels (Appendix A). The intensity of EGFP-Dcp2 puncta in cells overexpressing Nst1_∆PD_ was similar to that in cells overexpressing Nst1_∆APD_, which did not generate any concentrated EGFP-Dcp2 signals (Figure 3C and Figure 4C). These observations suggest that the PD is less correlated with self-condensation and may play a specific role in recruiting other PB components. Deleting the PD in Nst1_CTD_ also canceled out Nst1_CTD_ overexpression-induced EGFP-Dcp2 accumulation, supporting the role of PD in EGFP-Dcp2 condensation (Figure 4C,D).

The effect of the APD on Nst1 self-condensation was confirmed by Nst1_∆PD∆APD_ overexpression. We observed that GFP-tagged Nst1_∆PD∆APD_ was mainly dispersed in the cytoplasm as GFP-tagged Nst1_∆APD_, while overexpressed GFP-tagged Nst1_∆PD_ was observed as clear puncta (Figure 4A). Van Steensel’s CCF between GFP-tagged Nst1_∆PD∆APD_ and Dcp2-mKate2 also reflected that the double deletion of the PD and APD reduced the intrinsic self-condensation potential of overexpressed Nst1 to be co-localized with the PB marker (Figure 4B). These data confirm that the APD is responsible for Nst1 self-condensation. As expected, endogenous EGFP-Dcp2 condensates did not accumulate in cells overexpressing Nst1_∆PD∆APD_ (Figure 4C,D).

### 2.5. Dcp2 Condensation Induced by Nst1 PD Overexpression Is Independent of Free Ribosomal Influx

Observations of endogenous EGFP-Dcp2 in cells overexpressing various Nst1 mutants revealed that the APD and PD are largely responsible for self-condensation and inducing Dcp2 condensation, respectively. Since RNA functions as a scaffold for protein condensation via LLPS [12], we examined whether the PD in Dcp2 condensation is mediated by polysome RNA influx. We investigated EGFP-Dcp2 puncta induced by overexpression of each Nst1 domain deletion mutant after cycloheximide (CHX) treatment. PB formation induced by stress relies on an increase in non-translating mRNA concentration [12]. CHX completely disassembled the endogenous PBs formed during glucose deficiency, which inhibited translation elongation and resulted in a reduction in non-translating RNA [11,12]. Thus, protein-induced PB accumulation can be verified because CHX eliminated RNA-derived PBs.

Nst1, Nst1_∆PD_, Nst1_CTD_, and Nst1_CTD∆PD_ were overexpressed in cells with EGFP-tagged chromosomal *DCP2*, and EGFP-Dcp2 was observed after treating cells with 100 μg/mL CHX for 10 min [12]. Consistent with our previous report, Nst1 overexpression maintained EGFP-Dcp2 condensates in the presence of CHX, whereas the EGFP-Dcp2 puncta completely disappeared in the vector control (Figure 5A). We also observed that Nst1_CTD_ overexpression maintained EGFP-Dcp2 condensates in the presence of CHX (Figure 5A). The maximal intensity of EGFP-Dcp2 puncta generated by each domain deletion mutant was measured and plotted on the y-axis in the puncta quantification analysis shown in Figure 5A (Figure 5B). Similar to the results shown in Figure 4, the maximal intensity of EGFP-Dcp2 puncta accumulated by Nst1_∆PD_ overexpression was significantly decreased compared to that of full-length Nst1 overexpression and was similar to the vector-only control (Figure 5B). The maximal intensity of the EGFP-Dcp2 puncta accumulated by Nst1_CTD∆PD_ declined, similar to Nst1_∆PD_ (Figure 5B). The ratio of cells with generated EGFP-Dcp2 puncta in the PD deletion mutants (Nst1_∆PD_ and Nst1_CTD∆PD_) was dramatically decreased compared to that in Nst1 and Nst1_CTD_ (Figure 5C). These observations strongly support the implication that the PD is responsible for inducing the condensation of other PB components.

### 2.6. The Nst1 PD Serves as a Binding Hub, Mediating the Condensation of other PB Components

Edc3 is a PB scaffold protein in *S. cerevisiae* [11,54]. The ∆*edc3lsm4*∆C mutant could not induce EGFP-Dcp2 condensates independent of RNA influx, indicating that Edc3 is a critical component in PB generation. In our previous study, EGFP-Dcp2 condensation driven by Nst1 overexpression was suppressed in ∆*edc3 lsm4*∆C mutant cells, suggesting a functional relationship between Nst1 and Edc3 in condensate formation. The EGFP-tagged *EDC3* strain was transformed with the same Nst1 deletion mutant clones tested in Figure 5 to determine whether the Nst1 PD is also responsible for EGFP-Edc3 condensation. We then treated these cells with 100 μg/mL CHX after galactose induction to examine whether Nst1 PD-mediated Edc3 condensation is independent of polysome RNA influx. Microscopic observations revealed that the puncta of EGFP-Dcp2 and EGFP-Edc3, induced by Nst1 overexpression, behaved analogously. In the presence of CHX, EGFP-Edc3 condensation was highly decreased in cells overexpressing Nst1_∆PD_ and Nst1_CTD∆PD_ compared to that in cells overexpressing Nst1 and Nst1_CTD_ (Figure 6A). Nst1_∆PD_ overexpression did not induce EGFP-Edc3 puncta (Figure 6A), although GFP-tagged Nst1_∆PD_ overexpression resulted in its bright puncta (Figure 4B). The pattern of EGFP-Edc3 puncta generated by the overexpression of diverse Nst1 deletion mutants was similar to the pattern of EGFP-Dcp2, both in the maximal intensity and the ratio of puncta-generating cells (Figure 6B). The ratio of cells with generated EGFP-Edc3 puncta in the PD deletion mutants (Nst1_∆PD_ and Nst1_CTD∆PD_) was dramatically decreased compared to that in Nst1 and Nst1_CTD_ (Figure 6C). Endogenous EGFP-Edc3 expression levels of each Nst1 domain deletion mutant monitored by Western blot analysis showed that altering EGFP-Edc3 expression levels did not induce EGFP-Edc3 puncta reduction in cells overexpressing PD deletion mutants (Appendix A). These analyses strongly suggested that the PD is responsible for recruiting Edc3 as well as Dcp2.

The EGFP-tagged *DHH1* and *XRN1* strains were transformed with Nst1 and Nst1_∆PD_ and treated with 100 μg/mL CHX for 10 min after galactose induction to verify whether the PD recruits other PB components. Overexpression of PD deletion mutants (Nst1_∆PD_ and Nst1_CTD∆PD_) generated fewer Dhh1 and Xrn1 puncta than wild-type CHX-treated Nst1 overexpressing cells (Figure 6D,E). Overall, overexpression of the PD deletion mutants (Nst1_∆PD_ and Nst1_CTD∆PD_) reduced the condensation of known PB components, suggesting that the Nst1 PD interacts with PB components independent of polysome RNA influx.

## 3. Discussion

Understanding the syntax of biomolecular condensation is key to understanding the molecular dynamics of cells. RNA is a powerful scaffold, and the RNA-binding moiety of scaffold proteins is expected to be crucial in biomolecular condensation. However, the protein scaffolds responsible for condensation need to be investigated further. The sequence properties of various scaffold proteins in condensates, such as the low complexity domains (LCDs) of poly-Q or RGG and the polyampholytic region of charged amino acids (lysine or arginine), may be critical for biomolecular condensation [50,55]. Further, scaffold proteins that specifically function in a particular condensation generally have oligomerizing properties and IDRs. A study on the PB component, Lsm4, in budding yeast found that *GAL*-induced Lsm4 overexpression drives self-condensation [27]. Lsm4 is a representative PB component, with a prion-like domain (PrD, poly-Q motif) in its C-terminal region. Although CHX dissipated the stress-responsive endogenous Lsm4-GFP puncta, it did not disperse the bright clear puncta generated by the *GAL*-induced GFP-Lsm4, implying that the physical properties of the puncta induced by overexpressed Lsm4 were not identical to the stress-derived endogenous PBs. However, these observations indicate that Lsm4 has strong self-oligomerizing potential despite the unidentical physical properties of the puncta upon overexpression with native PBs. Similarly, overexpressed GFP-Edc3 appeared as bright clear puncta not dissipated by CHX, supporting previous reports that Edc3 harbors the Yjef-N domain, which induces Edc3 self-oligomerization [56].

We previously found that Nst1 significantly accumulated puncta in the stationary phase. We also reported that GFP-tagged Nst1 overexpression using a *GAL*-inducible promoter yielded condensates of round puncta (Figure 2A) and drove the accumulation of other PB components. These data strongly suggest that Nst1 has the potential to self-condensate and recruit other PB components to condense. CHX did not dissipate the overexpressed GFP-Nst1-generated bright clear puncta, suggesting that Nst1 has a sequence element that induces self-oligomerization similar to Lsm4 and Edc3, although Nst1 does not have a recognizable PrD. Nst1 is a large protein consisting of 1240 amino acids, including diverse sequence elements, a presumptive IDR, and aggregation-prone regions, as predicted by several programs (Figure 1). In this study, we attempted to elucidate the functional sequence elements of Nst1 for its self-condensation and accumulation of other PB component/s by examining GFP-tagged Nst1 domain deletion mutants upon overexpression.

### 3.1. The Nst1 C-Terminus Is Necessary and Sufficient for Self-Condensation, While the N-Terminus Has an Auxiliary Role in Recruiting other PB Components

Overexpression of GFP-tagged Nst1_NTD_ (residues 1–429) did not generate any self-condensation, whereas CTD (residues 430–1240) overexpression was sufficient for self-condensation, indicating that the oligomerizing domain is present in the Nst1 C-terminus. Nst1 condensation induced EGFP-Dcp2 condensation. The Nst1 domain, serving as a platform to interact with Dcp2, is essential for EGFP-Dcp2 condensation. In the analysis of Dcp2 condensates in cells overexpressing different Nst1 domain deletion mutants, CTD overexpression induced less Dcp2 accumulation than full-length Nst1. GFP-tagged Nst1_CTD_ formed bright clear puncta with a similar intensity to the GFP-tagged full-length Nst1 upon overexpression. Consistently, cells overexpressing the NTD did not seem to produce any EGFP-Dcp2 puncta compared to the cells overexpressing full-length Nst1 (Figure 2C,D), but instead showed reduced EGFP-Dcp2 puncta in comparison with the vector-only control. These observations can be explained by the recent LLPS mechanism suggested by the Brangwynne group, in which node capping could reduce interactor condensation [25]. By functioning as a Dcp2 node capper, the Nst1 NTD may directly or indirectly interact with Dcp2 to cover the Dcp2 node, resulting in Dcp2 condensation inhibition. These data imply that the overexpressed Nst1 NTD does not function in Nst1 self-condensation, but it may support Dcp2 recruitment to PB-associated condensates in full-length Nst1-overexpressing cells.

### 3.2. The Aggregation-Prone Region May Be Associated with Inducing Nst1 Condensates with Liquid-like Properties

We attempted to identify a specific region in the CTD that is responsible for self-condensation. According to Nst1 sequence-based predictions, the most aggregation-prone region consisted of hydrophobic amino acid residues in the APD. Although the precise link between aggregation propensity and LLPS remains unclear, the degree of aggregation propensity is likely correlated with many types of condensation, such as LLPS [57]. Among the several Nst1 domain deletion mutants constructed, the APD deletion mutant (Nst1_∆APD_) was the most powerful suppressor of Nst1 condensation driven by overexpression. Overexpressed GFP-tagged APD mutants, such as Nst1_∆APD_, Nst1_CTD∆APD_, and Nst1_∆PD∆APD_, had significantly decreased puncta and were dispersed in the cytoplasm (Figure 3A), suggesting that the APD in the C-terminus is critical for Nst1 self-condensation. Conversely, APD-overexpressing cells did not show any EGFP-Dcp2 condensates, although condensation of EGFP-Dcp2 in cells overexpressing Nst1_∆APD_ was alleviated compared to cells overexpressing full-length Nst1 (Figure 3C,D). These data suggest that APD is necessary but insufficient for Nst1 or Nst1_CTD_ self-condensation. The insufficiency of the APD for self-condensation could explain the importance of context and promiscuous interactions in protein condensation by LLPS [26,27,34,53]. These observations imply that although we could obtain clues on the sequence elements involved in Nst1 self-condensation by deleting each element, removing a domain may damage the unique sequence pattern of full-length Nst1 for condensation.

### 3.3. The Polyampholyte Region May Be Involved in Molecular Condensation as a Platform for Multivalent Protein–Protein Interactions Independent of RNA Influx

Polyampholytes containing a significant proportion (>35%) of positively and negatively charged residues are present in 75% of intrinsically disordered proteins [48,55]. In the analyses of EGFP-Dcp2 puncta formed in cells overexpressing Nst1 domain deletion mutants, the Nst1 PD was responsible for recruiting other PB components. The cells overexpressing the PD deletion mutants showed significantly decreased Dcp2 puncta compared to the full length and CTD of Nst1. However, the overexpressed GFP-tagged PD deletion mutants strongly induced self-condensation, similar to overexpressed GFP-tagged Nst1 and Nst1_CTD_ (Figure 4A). The decreased EGFP-Dcp2 phenotype in the overexpressed PD deletion mutants was more clearly observed after treatment with CHX (Figure 5A,B).

Other PB markers, such as Edc3, Dhh1, and Xrn1, showed similar accumulation patterns to Dcp2 in CHX-treated cells overexpressing PD deletion mutants. These observations clarified that PD does not affect Nst1 oligomerization but significantly contributes to the recruitment of other PB components, namely Dcp2, Dhh1, Xrn1, and Edc3.

To assess the physical interaction of Nst1 and Nst1_ΔPD_ with PB components upon biomolecular condensation, we first checked the physical interactions of Nst1 with other essential PB constituents via a co-immunoprecipitation (Co-IP) assay of the 6hemagglutinin (HA)-tagged Nst1 with 9Myc-tagged Dcp2/Edc3/Dhh1/Ccr4 in the log phase cells, but could not observe any interaction (data not shown). We also tried to identify whether the interaction between the overexpressed full-length Nst1, Nst1_ΔPD_, Nst1_CTD_ Nst1_CTDΔPD_ and Dcp2 is biochemically detected. Co-IP of the overexpressed Nst1 and Nst1 domain deletion mutants with 9Myc-tagged Dcp2 was performed. Unfortunately, we could not detect any direct physical interaction between them (Appendix A), although microscopic examinations provided evidence of Dcp2, Edc3, Dhh1, and Xrn1 condensation and co-localization with Nst1 when Nst1 was overexpressed (Figure 5 and Figure 6). These results imply that the interactions between Nst1 and known PB components were not strong enough to be detected biochemically but were sufficient to induce promiscuous interactions. Considering that polyampholyte regions are among the most frequently occurring IDRs in nature, our findings provide insights into the roles of polyampholyte multivalency in interacting with other PB constituents. Further studies are needed to understand the molecular mechanism by which the Nst1 polyampholyte recruits other PB components to form condensates.

This study on Nst1 domain deletion mutants further improves our understanding of the sequence elements with high aggregation propensity and the polyampholyte region in unstructured proteins involved in the self-condensation and condensation of PB components with liquid-like properties.

## 4. Materials and Methods

### 4.1. Yeast strains, Plasmids, and Cultures

Table 1 lists the *S. cerevisiae* strains and genotypes used in this study. The strains were constructed on the BY4741 or w303a wild-type background by integrating templates from the polymerase chain reaction (PCR) toolbox at the 3′ end of each reading frame in each endogenous locus through PCR-based homologous recombination [58]. We used PCR of the integrated locus and Western blotting to verify all the constructed strains.

All plasmids used in this study were constructed in *pMW20(U)-**P_GAL_-GFP* or *pMW20(U)-**P_GAL_*, as Table 2 describes. Nst1 domain deletion mutant clones were generated using the PCR-mediated deletion method [59] and confirmed by sequencing.

Yeast strains were cultured in YPAD or synthetic complete (SC) media containing 2% glucose. Glucose deprivation was induced in the SC medium without glucose. Yeast cells were cultured at 25 °C to an optical density of 600 nm (OD_600_) ≤ 0.5 for logarithmic phase growth. Cells in the logarithmic phase were primarily cultured in SC-U + 2% glucose media to an OD_600_ of 0.5 and harvested to induce overexpression under the *GAL*. The cells were washed three times with Sc-U + 2% raffinose + 0.1% glucose medium, diluted to half of its concentration, and cultured for an additional 3 h in Sc-U + 2% raffinose + 0.1% glucose. Then, 20% galactose stock was added to the culture to adjust the final galactose concentration to 2%, and the cells were further incubated for 3 h for induction before collection.

### 4.2. 1,6-Hexanediol and CHX Treatments and Western Blots

Yeast *GAL* promoter induction was performed as described above. For CHX treatment, *GAL*-induced cells were incubated with 100 μg/mL CHX for 10 min. For 1,6-hexanediol treatment, the *GAL*-induced cells were washed three times with a medium containing 10% 1,6-hexanediol and incubated for 30 min. Western blotting was performed as described by Choi and Song [40] with anti-EGFP antibody (600-101-215 Rockland, Limerick, PA, USA) and anti-Tub1 (T5168, Sigma, St. Louis, MO, USA) positive controls. HRP-conjugated anti-goat (705-035-003, Jackson Immune Research, PA, USA) and anti-mouse (sc-2005, Santa Cruz Biotechnology, Dallas, TX, USA) antibodies were used as secondary antibodies to detect EGFP and anti-Tub1, respectively.

### 4.3. Nst1 Structure and Domain Predictions Based on the Sequence

Structural prediction of Nst1 was performed using GalaxyWEB [60]. The IUPRED2A, PONDR, and DISOPRED3 [42,43,61,62] algorithms, which predict IDRs, were used to analyze the disorder properties in the Nst1 sequence. AGGRESCAN, Tango, and PASTA 2.0 were used to predict aggregation-prone regions in the Nst1 sequence [45,46,63,64,65]. Domain deletion mutants of Nst1 were projected in the Das–Pappu phase diagram at http://pappulab.wustl.edu/CIDER/analysis/, (accessed on 6 April 2022).

### 4.4. Wide-Field Fluorescence Microscopy of Yeast Cells and Image Analysis

Fluorescence-labeled proteins were visualized using an Axioplan2 microscope (Carl Zeiss, Jena, Germany) with a 100× Plan-Neofluar oil immersion objective. Images were acquired using an Axiocam CCD camera and Axio Vision software (Carl Zeiss). The same culture conditions, exposure times, and fluorescence intensities were applied to all the strains observed in this study to compare the degree of puncta intensity. Images were analyzed as described below.

For colocalization analysis to deduce van Steensel’s CCF, fluorescent images obtained through green or red channels were analyzed with the plugin JACoP v4.0 analysis tool using FIJI (Image J). CCFs were calculated and presented as a bell-shaped plot.

All images obtained were measured and analyzed with the same optics, filters, and zoom settings throughout the study using FIJI (ImageJ) to quantify PB condensation.

The pixel intensity of each GFP signal in the cells was scaled from 1 to 255 to investigate the intensity of puncta per cell. Pixels of the top 0.05% signal intensities of each strain were segmented to perform particle analysis and determine the individual punctum strength to deduce the maximum intensity value of each punctum. The highest pixel value of each punctum was presented. The total number of cells, the number of cells with puncta, and the ratio of cells with puncta were calculated to analyze the puncta generated.

### 4.5. Statistical Analysis

Detailed statistics, including the mean values and standard deviations, are indicated in each figure legend. Statistical analyses were performed using GraphPad Prism 6 (GraphPad Software, Inc., La Jolla, CA, USA). A t-test was used to assess statistically significant differences. *p* < 0.05 (*), *p* < 0.01 (**), *p* < 0.001 (***), and *p* < 0.0001 (****) indicate statistical significance compared with the control. *p* > 0.05 indicates statistical non-significance (n.s.).

## Figures and Tables

**Figure 1 ijms-23-07380-f001:**
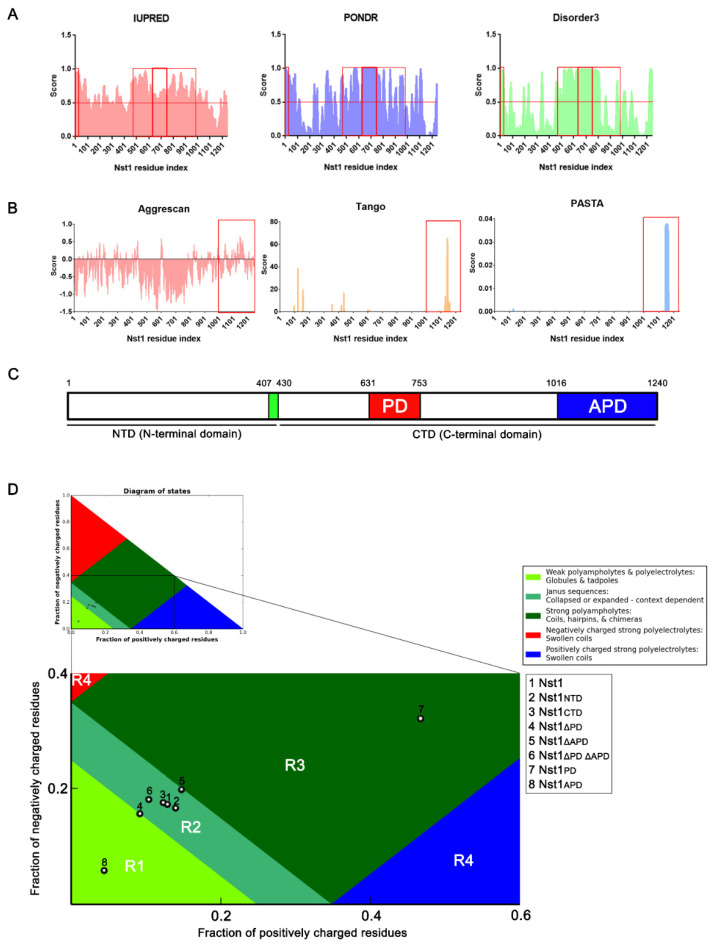
Nst1 contains a polyampholyte domain (PD) and a predicted aggregation-prone region. (**A**) Predicting the Nst1 intrinsically disordered regions (IDRs). IUPRED2A (red) [41], PONDR (blue) [42], and DISOPRED3 (green) [43] algorithms were used for the prediction. Disorder scores were calculated and presented. Scores exceeding the 0.5 threshold indicate the amino acid residues in the Nst1 disordered regions. The disordered regions with scores >0.5 in all three algorithms used are identified as IDRs and highlighted in red-lined boxes. A length threshold for the disordered regions is also set to >30 residues [44]. The PD in predicted disordered regions is marked with thick red-lined boxes. PD corresponding residues are labeled in Supplemental Appendix A. (**B**) Predicting the Nst1 aggregation-prone regions. The regions with high aggregation propensities were calculated using AGGRESCAN (purple) [45], Tango (orange) [46], and PASTA 2.0 (blue) [47] algorithms. Amino acid residues with scores greater than the threshold value in all three algorithms were aggregation-prone and marked with a red box in the Nst1 sequence. Corresponding residues are labeled in Supplemental Appendix A. (**C**) A diagram of Nst1 with domain architectures predicted by (**A**,**B**), and GalaxyWEB. Each color in the schematic corresponds to a particular domain in the sequence. (**D**) Das–Pappu diagrams of the full-length Nst1 and its domain deletion mutants. The full-length Nst1 and its various domain deletion mutants are numbered in the box: 1. Nst1 (residues 1–1240), 2. Nst1_NTD_ (N-terminal domain (NTD) Nst1 residues 1–429), 3. Nst1_CTD_ (C-terminal domain (CTD) Nst1 residues 430–1240), 4. Nst1_ΔPD_ (residues 1–630 and 753–1240), 5. Nst1_ΔAPD_ (residues 1–1015), 6. Nst1_ΔPDΔAPD_ (residues 1–630 and 753–1015), 7. Nst1_PD_ (residues 631–752), and 8. Nst1_APD_ (residues 1016–1240). The *x*- and *y*-axes represent the fraction of positively and negatively charged residues, respectively. The four zones (R1–R4) of the diagram are colored in bright green (R1), emerald (R2), forest (R3), and red/blue (R4), respectively. The physicochemical properties of each colored zone are explained in the inset. The numbers for the full-length Nst1 and its domain deletion mutants are assigned to a corresponding region from R1–R4 with a circle.

**Figure 2 ijms-23-07380-f002:**
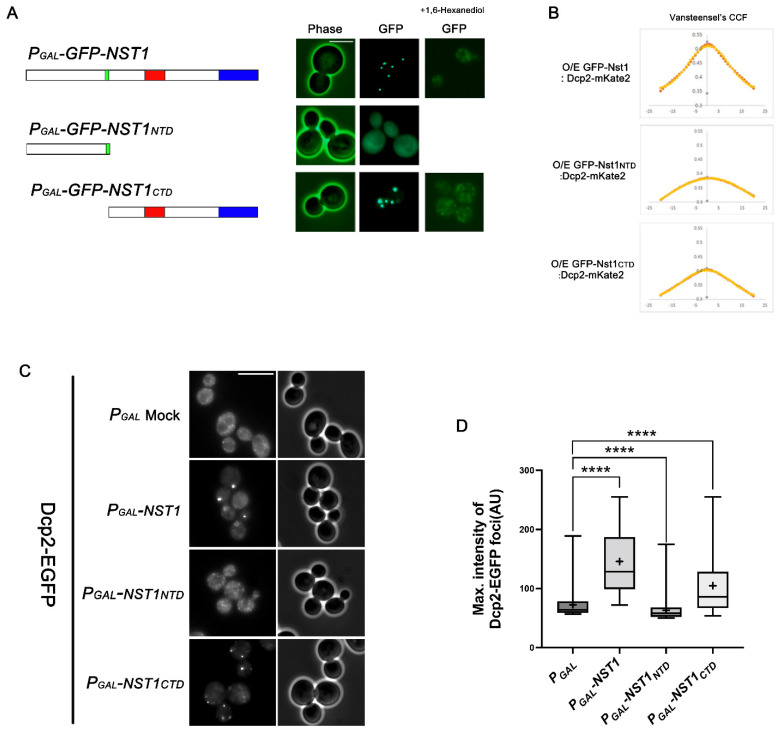
The Nst1 CTD is sufficient for Nst1 self-condensation. (**A**) Fluorescence microscopy of the cells overexpressing full-length enhanced green fluorescent protein (EGFP)-tagged Nst1 and the NTD (Nst1_NTD_, 1429) and CTD (Nst1_CTD_, 430–1240) of Nst1. Overexpression of each EGFP-tagged Nst1 deletion mutant was induced in wildtype cells, then observed before and after 1,6-hexanediol treatment. Scale bar: 5 μm. Schematic diagrams of the designed Nst1 domain deletion mutants are shown on the left. (**B**) The van Steensel’s crosscorrelation coefficients (CCFs) between each overexpressed Nst1 deletion mutant used in (**A**). The endogenous mRNA decapping protein 2 (Dcp2)-mKate2 signals were analyzed and presented. Overexpression of each EGFP-tagged Nst1 domain deletion mutant was induced in wild-type cells whose chromosomal *DCP2* was tagged with mKate2. Each Nst1 domain deletion mutant (n = total observed cell number): *P_GAL_*-*GFP*-*NST1* (n = 257), *P_GAL_*-*GFP*-*NST1_NTD_* (n = 158), and *P_GAL_*-*GFP*-*NST1_CTD_* (n = 161). All images were analyzed by FIJI (https://imagej.net/Fiji, accessed on 9 August 2020). (**C**,**D**) Each Nst1 deletion mutant was overexpressed in the wild-type cells with EGFP-tagged *DCP2* (YSK3485). (**C**) Fluorescence microscopy of cells expressing endogenous EGFP-Dcp2 that overexpress the NTD (1–429), CTD (430–1240), and full-length of Nst1. Scale bar: 10 μm. (**D**) Quantification of the endogenous EGFP-Dcp2 puncta analysis of (**C**). The pixels of the top 0.1% EGFP-Dcp2 signal intensities were segmented for puncta analysis. The maximal intensity of each segmented punctum was plotted. ‘+’ in the boxplot indicates the mean value of maximal intensities of foci. Each Nst1 domain deletion mutant (n = total observed cell number): *P_GAL_* vector—only control (n = 218), *P_GAL_*-*NST1* (n = 307), *P_GAL_*-*NST1_NTD_* (n = 333), and *P_GAL_*-*NST1_CTD_* (n = 300). All measurements and analyses were performed by FIJI (https://imagej.net/Fiji, accessed on 31 March 2022). Statistical significance was determined by a Mann–Whitney test **** *p* < 0.0001).

**Figure 3 ijms-23-07380-f003:**
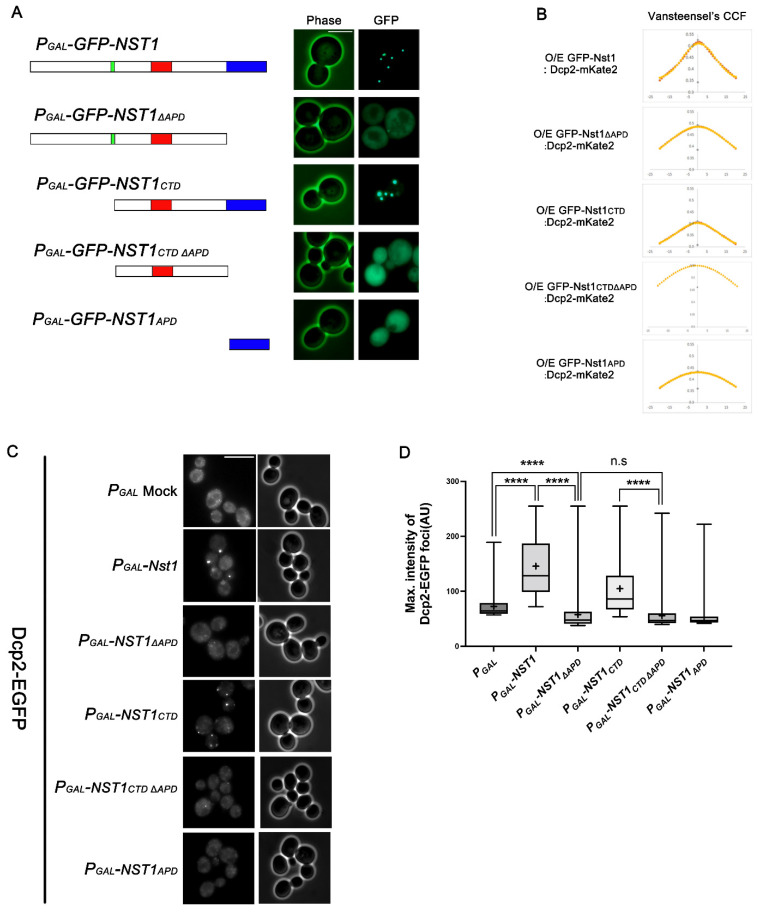
The aggregation-prone domain (APD) in the Nst1_CTD_ is insufficient but crucial for inducing Nst1 self-condensation. (**A**) Fluorescence microscopy of cells overexpressing EGFP-tagged full-length Nst1, Nst1_ΔAPD_ (residues 1–1015), Nst1_CTD_ (C-terminal Nst1 residues 430–1240), Nst1_CTDΔAPD_ (residues 431–1015), and Nst1_APD_ (residues 1016–1240). Schematic diagrams of the designed Nst1 domain deletion mutants are shown on the left. Overexpression of each EGFP-tagged Nst1 domain deletion mutant was induced in wild-type cells, then observed. Scale bar: 5 μm. (**B**) The van Steensel’s CCFs between each overexpressed Nst1 domain deletion mutant used in (**A**) and the endogenous Dcp2-mKate2 signals were analyzed and presented. Overexpression of each EGFP-tagged Nst1 domain deletion mutant was induced in wild-type cells whose chromosomal *DCP2* was tagged with mKate2. Each Nst1 domain deletion mutant (n = total observed cell number): *P_GAL_-GFP-NST1* (n = 257), *P_GAL_-GFP-NST1**_ΔAPD_* (n = 387), *P_GAL_-GFP-NST1_CTD_* (n = 161), *P_GAL_-GFP-NST1_CTD_**_ΔAPD_* (n = 191), and *P_GAL_-GFP-NST1_APD_* (n = 167). All images were analyzed by FIJI (https://imagej.net/Fiji, accessed on 9 August 2020). (**C**,**D**) Each Nst1 domain deletion mutant was overexpressed in the wild-type cells with EGFP-tagged *DCP2* (YSK3485). (**C**) Fluorescence microscopy of endogenous EGFP-Dcp2-tagged cells overexpressing full-length Nst1, Nst1_ΔAPD_, Nst1_CTD_, Nst1 _CTDΔAPD_, and Nst1_APD_. Scale bar: 10 μm. (**D**) Quantification of the endogenous EGFP-Dcp2 puncta of (**B**). The pixels of the top 0.1% EGFP-Dcp2 signal intensities were segmented for puncta analysis. The maximal intensity of each segmented punctum was plotted. ‘+’ in the boxplot indicates the mean value of maximal intensities of foci. Each Nst1 domain deletion mutant (n = total observed cell number): *P_GAL_* vector-only control (n = 218), *P_GAL_-NST1* (n = 307), *P_GAL_-NST1**_ΔAPD_* (n = 337), *P_GAL_-NST1_CTD_* (n = 300), *P_GAL_-NST1_CTD_**_ΔAPD_* (n = 260), and *P_GAL_-NST1_APD_* (n = 261). All measurements and analyses were performed by FIJI (https://imagej.net/Fiji, accessed on 31 March 2022). Statistical significance was determined by a Mann–Whitney test (**** *p* < 0.0001).

**Figure 4 ijms-23-07380-f004:**
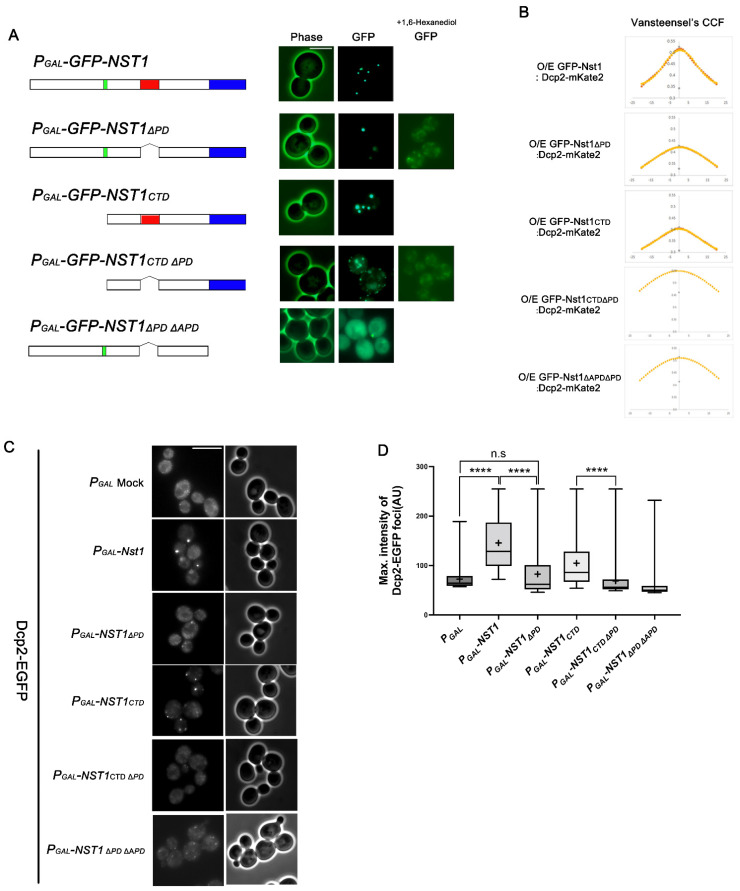
The polyampholyte region is not critical for Nst1 self-condensation but is responsible for inducing Dcp2 condensation. (**A**) Fluorescence microscopy of cells overexpressing EGFP-tagged full-length Nst1, Nst1_ΔPD_ (residues 1–630 and 753–1240), Nst1_CTD_ (C-terminal Nst1 residues 430–1240), Nst1_CTDΔPD_ (residues 431–630 and 753–1240), and Nst1_ΔPDΔAPD_ (residues 1–630 and 753–1015). Overexpression of each EGFP-tagged Nst1 domain deletion mutant was induced in wild-type cells, then observed before and after 1,6-hexanediol treatment. Scale bar: 5 μm. (**B**) The van Steensel’s CCFs between each overexpressed Nst1 domain deletion mutant used in (**A**) and the endogenous Dcp2-mKate2 signals were analyzed and presented. Overexpression of each EGFP-tagged Nst1 domain deletion mutant was induced in wild-type cells whose chromosomal *DCP2* was tagged with mKate2. Each Nst1 domain deletion mutant (n = total observed cell number): *P_GAL_-GFP-NST1* (n = 257), *P_GAL_-GFP-NST1**_ΔPD_* (n = 277), *P_GAL_-GFP-NST1_CTD_* (n = 161), *P_GAL_-GFP-NST1_CTD_**_ΔPD_* (n = 199), and *P_GAL_-GFP-NST1**_ΔAPDΔPD_* (n = 198). All images were analyzed by FIJI (https://imagej.net/Fiji, accessed on 9 August 2020). (**C**,**D**) Each Nst1 domain deletion mutant was overexpressed in the wild-type cells with EGFP-tagged *DCP2* (YSK3485). Schematic diagrams of the designed Nst1 domain deletion mutants are shown on the left. (**C**) Fluorescence microscopy of endogenous EGFP-Dcp2-tagged cells overexpressing full-length Nst1, Nst1_ΔPD_ (residues 1–630 and 753–1240), Nst1_CTD_ (C-terminal Nst1 residues 430–1240), Nst1_CTDΔPD_ (residues 431–630 and 753–1240), and Nst1_ΔPDΔAPD_ (residues 1–630 and 753–1015). Scale bar: 10 μm. (**D**) Quantification of the endogenous EGFP-Dcp2 puncta of (**C**). The pixels of the top 0.1% EGFP-Dcp2 signal intensities were segmented for puncta analysis. The maximal intensity of each segmented punctum was plotted. ‘+’ in the boxplot indicates the mean value of maximal intensities of foci. Each Nst1 domain deletion mutant (n = total observed cell number): *P_GAL_* vector-only control (n = 218), *P_GAL_-NST1* (n = 307), *P_GAL_-NST1**_ΔPD_* (n = 284), *P_GAL_-NST1_CTD_* (n = 300), *P_GAL_-NST1_CTD_**_ΔPD_* (n = 200), and *P_GAL_-NST1**_ΔPDΔAPD_* (n = 260). All measurements and analyses were performed by FIJI (https://imagej.net/Fiji, accessed on 31 March 2022). Statistical significance was determined by a Mann–Whitney test (**** *p* < 0.0001).

**Figure 5 ijms-23-07380-f005:**
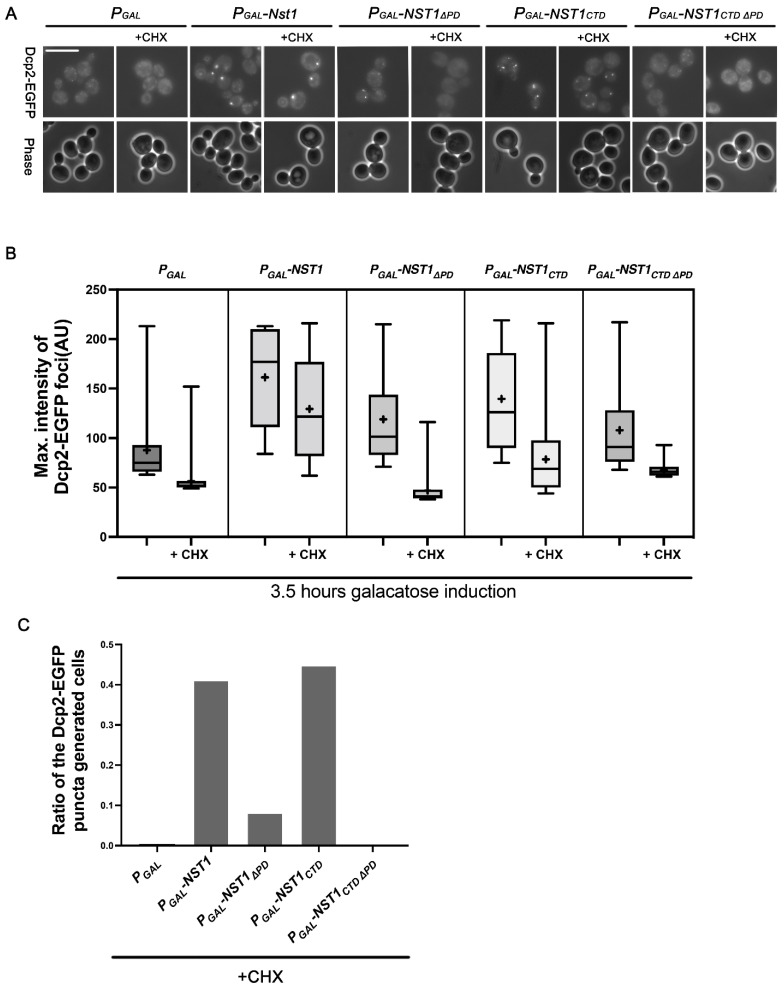
The Nst1 polyampholyte region interacts with a PB component Dcp2 independent of the free ribosomal RNA influx. (**A**–**C**) In the wild-type strain whose chromosomal *DCP2* was tagged with EGFP, endogenous EGFP-Dcp2 was observed after the overexpression of full-length Nst1, Nst1_ΔPD_ (residues 1–630 and 753–1240), Nst1_CTD_ (C-terminal Nst1 residues 430–1240), and Nst1_CTD__ΔPD_ (residues 431–630 and 753–1240). In the cells overexpressing each Nst1 domain deletion mutant, endogenous Dcp2 was observed before and after the 10 min 100 μg/mL cycloheximdie (CHX) treatment. (**A**) Fluorescence microscopy of EGFP-Dcp2 in the cells overexpressing each mutant before and after the 10 min 100 μg/mL CHX treatment. Scale bar: 10 μm. (**B**) Quantification of EGFP-Dcp2 puncta shown in (**A**). The pixels of the top 0.1% signal intensities were segmented and analyzed. The maximal value of each punctum was plotted. ‘+’ in the boxplot indicates the mean value of maximal intensities of foci. All measurements and analyses were performed by FIJI (https://imagej.net/Fiji, accessed on 31 March 2022) (**C**) The ratio of cells producing EGFP-Dcp2 puncta to the total cells by the overexpression of each Nst1 mutant (Nst1 domain deletion mutant (n = total observed cell number): *P_GAL_* (vector only, n = 260), *P_GAL_-NST1* (n = 312), *P_GAL_-NST1**_ΔPD_* (n = 247), *P_GAL_-NST1_CTD_*(n = 302), and *P_GAL_-NST1_CTD_**_ΔPD_*(n = 241)) in (**A**).

**Figure 6 ijms-23-07380-f006:**
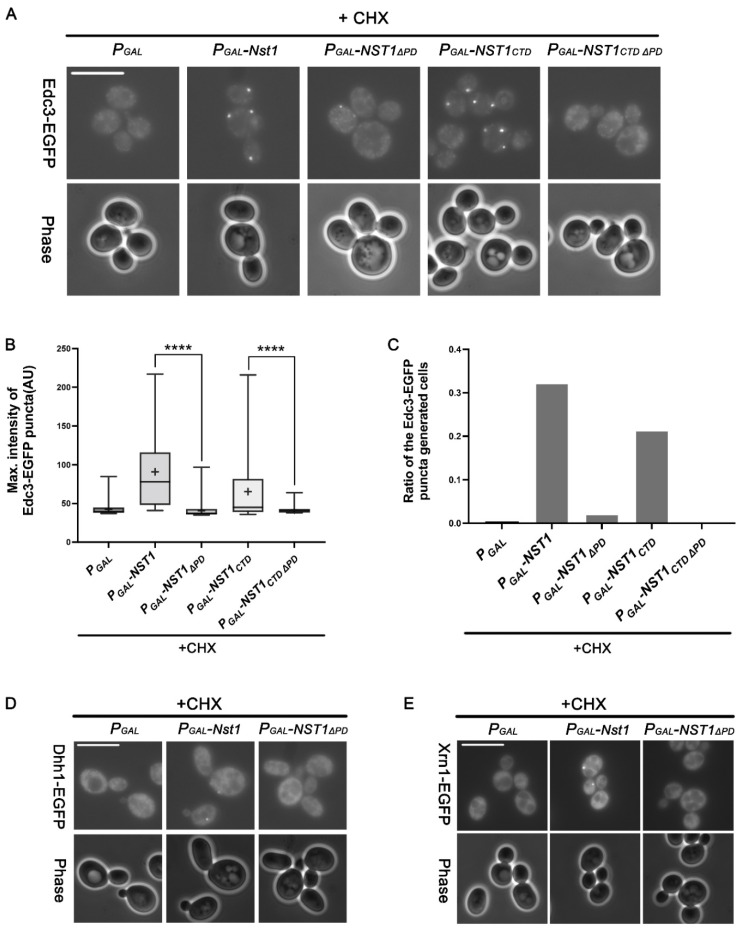
The Nst1 polyampholyte region functions as a binding hub for P-body (PB) components independent of the free ribosomal RNA influx. (**A**–**C**) In the wild-type strains whose chromosomal EDC3 was tagged with EGFP, the full-length Nst1, Nst1ΔPD (residues 1–630 and 753–1240), Nst1CTD (C-terminal Nst1 residues 430–1240), and Nst1CTDΔPD (residues 431–630 and 753–1240) were overexpressed. Endogenous enhancer of mRNA decapping 3 (Edc3) was observed before and after cells were treated with 100 μg/mL CHX for 10 min. (**A**) Fluorescence microscopy of EGFP-Edc3 in the cells overexpressing each mutant before and after the 10 min 100 μg/mL CHX treatment. Scale bar: 10 μm. (**B**) Quantification of the EGFP-Edc3 puncta shown in (**A**) with CHX. The pixels of the top 0.1% signal intensities were segmented as the EGFP-Edc3 puncta. The EGFP-Edc3 puncta generated were quantified, and the maximal value of each punctum was plotted. ‘+’ in the boxplot indicates the mean value of maximal intensities of foci. All measurements and analyses were performed by FIJI (https://imagej.net/Fiji, accessed on 31 March 2022) Statistical significance was determined by a Mann–Whitney test (**** *p* < 0.0001). (**C**) The ratio of cells producing EGFP-Edc3 puncta to the total cells overexpressing each Nst1 mutant (Nst1 domain deletion mutant in the presence of CHX (n = total observed cell number): PGAL (vector only, n = 233), PGAL-NST1 (n = 320), PGAL-NST1ΔPD (n = 200), PGAL-NST1CTD (n = 244), and PGAL-NST1CTDΔPD (n = 275). (**D**,**E**) In the wild-type strains with EGFP-tagged chromosomal DHH1 and XRN1, the overexpression of full-length Nst1 and Nst1ΔPD cells was induced, and then cells were treated with 100 μg/mL CHX for 10 min. Fluorescence microscopy of (**D**) EGFP-Dhh1 and (**E**) EGFP-Xrn1 in CHX-treated cells overexpressing full-length Nst1 and Nst1ΔPD. Scale bar: 10 μm. All images were measured and analyzed by FIJI (https://imagej.net/Fiji, accessed on 31 March 2022).

**Table 1 ijms-23-07380-t001:** The yeast strains used in this study.

Strain Name	Genotype	Source
YSK3485	*DCP2-EGFP:HIS3MX6 BY4741 MATa his3*Δ*1 leu2*Δ*0 met15*Δ*0 ura3*Δ*0*	This study
YSK3482	*XRN1-EGFP::HIS3MX6 BY4741 MATa his3*Δ*1 leu2*Δ*0 met15*Δ*0 ura3*Δ*0*	This study
YSK3484	*DHH1-EGFP:HIS3MX6 BY4741 MATa his3*Δ*1 leu2*Δ*0 met15*Δ*0 ura3*Δ*0*	This study
YSK3534	*EDC3-EGFP::HIS3MX6 BY4741 MATa his3*Δ*1 leu2*Δ*0 met15*Δ*0 ura3*Δ*0*	This study
YSK3578	*DCP2-mKate2-sphis5 BY4741 MATa his3*Δ*1 leu2*Δ*0 met15*Δ*0 ura3*Δ*0*	This study
YSK3483	*BY4741 MATa his3*Δ*1 leu2*Δ*0 met15*Δ*0 ura3*Δ*0 wild-type*	This study
YSK3592	*DCP2-9MYC::HIS3MX6 BY4741 MATa his3*Δ*1 leu2*Δ*0 met15*Δ*0 ura3*Δ*0*	This study

**Table 2 ijms-23-07380-t002:** The plasmids used in this study.

Plasmids
*pMW20-P_GAL_-GFP-NST*
*pMW20-P_GAL_-GFP-NST*Δ*430–1240*
*pMW20-P_GAL_-GFP-NST1*Δ*1–429*
*pMW20-P_GAL_-GFP-NST1*Δ*1–429* Δ*631–752*
*pMW20-P_GAL_-GFP-NST1*Δ*1–429* Δ*1016–1240*
*pMW20-P_GAL_-GFP-NST1*Δ*1–1015*
*pMW20-P_GAL_-GFP-NST1*Δ*631–752*
*pMW20-P_GAL_-GFP-NST1*Δ*631–752* Δ*1016–1240*
*pMW20-P_GAL_-GFP-NST1*Δ*1016–1240*
*pMW20-P_GAL_-GFP-NST1*Δ*1–1015*
*pMW20-P_GAL_-NST*
*pMW20-P_GAL_-NST*Δ*430–1240*
*pMW20-P_GAL_-NST1*Δ*1–429*
*pMW20-P_GAL_-NST1*Δ*1–429* Δ*631–752*
*pMW20-P_GAL_-NST1*Δ*1–429* Δ*1016–1240*
*pMW20-P_GAL_-NST1*Δ*631–752*
*pMW20-P_GAL_-NST1*Δ*631–752* Δ*1016–1240*
*pMW20-P_GAL_-NST1*Δ*1016–1240*
*pMW20-P_GAL_-NST1*Δ*1–429* Δ*631–752*

## Data Availability

Not applicable.

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
