# Peer review of "The Multivalent Polyampholyte Domain of Nst1, a P-Body-Associated Saccharomyces cerevisiae Protein, Provides a Platform for Interacting with P-Body Components"

_ijms, 2022, doi:10.3390/ijms23137380_

Round 1
Reviewer 1 Report
The reviewer’s comments.
The manuscript entitled "The multivalent polyampholyte domain of Nst1, a P-body-associated Saccharomyces cerevisiae protein, provides a platform for interacting with P-body components" contributed by Jeong Choi et al., shows that the polyampholyte domain (PD) the IDR in Nst1 functions as a hub domain interacting with other P-body components. The authors have employed a sophisticated system from Yeast to indicate critical interactions of the PD of IDR in Nst1 with Dcp2, Dcp3, and more components of P-body. Authors emphasize essential role of the PD in NST1 as the hub domain of interaction with other p-body components. This present a model to understand unsolved principle for constructing membraneless organelles. Regarding the data presented here, the reviewer would like to clarify one issue before publication of the manuscript as follows.
At Figure 4A, full-length GFP-NST1 and GFP-NST1 C-terminus domain have made condensates. At Figure 4C, the full-length NST1 and NST1 CTD induced Dcp2 condensates. However, NST-CTD deleted with PD lost ability to induce condensates, suggesting that the PD plays a role in its association with Dcp2. Then,the reviewer would like to see these interactions between NST1 and Dcp2 in a biochemical detection system like pull-down assay using agarose beads equipped a Tag for grasp of GFP-NST1 or Dcp2-EGFP. The fragment of GFP-NST1 CTD might bind Dcp2-EGFP. Such kinds of experiments confirm the results with a biochemical view point. This should be beneficial for the manuscript.
Author Response
- As we mentioned in 3.3 section of the Discussion, we performed Co-IP assay to examine whether the physical interaction between Nst1 and Dcp2/Edc3/Dhh1/Ccr4 is detected endogenously. However, none of the interactions was identified biochemically, representing weakness of binding force in between Nst1 and other PB components. Considering that biomolecular phase separation arises based on weak multivalent interactions (David W. Sanders et al., 2020), we could hardly assume that the binding affinity between the full-length Nst1 and other PB components is strong enough to be detected with a biochemical method such as Co-IP, even when Nst1 is overexpressed. However, by following the reviewer suggestion, we had performed Co-IP between the overexpressed GFP-Nst1, GFP-Nst1DPD, Nst1CTD, Nst1CTDDPD, and chromosomally 9Myc-tagged Dcp2. We could not detect any physical interaction of overexpressed Nst1 domain deletion mutants with 9Myc-tagged Dcp2 by Co-IP analysis, as well as the overexpressed full-length Nst1, as shown in the Supplementary Figure S3 of the revised manuscript. These results confirm weak multivalent interactions between the overexpressed full-length Nst1 and Dcp2.

Reviewer 2 Report
In this work, Choi et al. performed a highly comprehensive study on the roles of each segment of the Nst1 in the condensate phase separation formation. They started from the bioinformatic analysis of the classifications of the Nst1 and decided to divide the Nst1 into several key segments illustrated in Figure 1. Then, they performed individual experiments on different Nst1 mutants to study the potential for forming the condensates. Finally, they were able to draw conclusions about the roles of the segments based on their well-designed experiments. Overall, the work was well performed, and the manuscript was written clearly. I enjoyed the reading and therefore would recommend its publication in this journal. I only have one minor comment for the authors to consider.
In Figure 1, Nst1 was classified into different segments, which were further to be studied experimentally. However, I am not very clear how these NTD, PD and APD were determined. For example, in Figure 1A, it appears to me that there are many other regions with scores> 0.5, in addition to the ones marked by the authors already. I would appreciate if the authors could elaborate on this point.
Author Response
- For all Nst1 domain designations, we considered a Nst1’s secondary structure prediction as well as the prediction tools for intrinsic disordered region (IDR) and aggregation-prone region (APR) presented in Fig. 1. When GalaxyWEB (http://galaxy.seoklab.org/structure prediction toolno globular structure but only some secondary structures were predicted as shown in Supplementary Figure 1 B and C of the revised manuscript. We designated each domain of Nst1, considering the secondary structures predicted in GalaxyWEB: Nst1 was divided into NTD and CTD, with a short unstructured region in-between (Supplementary Figure 1 B and C).
- In case of intrinsic disordered region (IDR) and aggregation-prone domain (APD), as you mentioned, many other regions also showed high scores in IDR and APR prediction tools. The regions with scores > 0.5 in all three algorithms used (IUPRED, PONDR, and Disorder 3) were identified as an IDR and highlighted in red. We also set a length threshold for the IDR to > 30 residues (Dunker et al., 2013). For example, in this regard, we did not mark residues 332-347 (less than 30 residues) as IDR in Fig. S1, although they exhibited scores > 0.5 in all three algorithms. We added the details of domain prediction in the legend of Figure 1 A and B of the revised manuscript. In addition, we considered the repeats of charged amino acids in polyampholyte region to assign polyampholyte domain (PD). As shown in Das-Pappu diagram of Fig. 1 D, the polyampholyte domain (PD) has abnormally high portion of charged residues. Moreover, the polyampholyte region was predicted as helical helices by GalaxyWEB and we designated the PD to minimize damages of other predicted secondary structures nearby. The consensus of aggregation-prone amyloidogenic regions among three different algorithms (Aggrescan, Tango, and PASTA) were colored in blue. The APD is the only region predicted as globular structure by GalaxyWEB and contains the aggregation-prone residues by all three prediction tools used.
